# Spinal Schwannomatosis Mimicking Metastatic Extramedullary Spinal Tumor

**DOI:** 10.3390/diagnostics13071254

**Published:** 2023-03-27

**Authors:** Idris Nurdillah, Iqbal Hussain Rizuana, Sharis Osman Syazarina

**Affiliations:** Department of Radiology, Faculty of Medicine, Universiti Kebangsaan Malaysia Medical Center, Jalan Yaakob Latif, Kuala Lumpur 56000, Malaysia

**Keywords:** intradural, extramedullary, schwannoma, hemangioma, metastases

## Abstract

Intradural extramedullary (IDEM) tumors are the most commonly observed intraspinal tumors, comprising over 60% of tumors found within the spinal canal, and the vast majority of these lesions are benign lesions. IDEM metastases are rare, but if they occur, they commonly manifest as leptomeningeal disease, secondary to drop lesions from intracranial metastases from adenocarcinomas of the lung, prostate cancer, breast cancer, melanoma, or rarely, as a result of lymphomas. The purely non-neurogenic origin of IDEM metastases is rare. Herein, we describe a patient with a previous history of treated colon cancer who presented with a progressive neurological deficit and whose imaging revealed multiple intradural, extramedullary and osseous lesions at the cervical and thoracolumbar spines. With the previous known primary and multiplicity of the lesions, an initial diagnosis of spinal metastasis was made, But it was proven to be schwannoma on histology. We emphasize the diagnostic dilemma in this case and the importance of detecting subtle imaging findings, which may be helpful to differentiate between metastatic disease and a second primary tumor.

The patient was a 72-year-old gentleman with underlying dyslipidemia. He had a history of ascending colon adenocarcinoma, diagnosed six years ago. He underwent a right hemicolectomy, completed chemoradiotherapy in the same year, and has been in disease remission since then. He presented to us with a two-week history of progressive bilateral upper and lower limb weakness, predominantly affecting the upper limbs more on the right side. Later, he developed difficulty walking and a deterioration of his gait due to weakness. However, he was still able to move all four limbs and did not complain of any sphincter dysfunction or constitutional symptoms.

Examination showed no evidence of neurocutaneous markers. Neurological evaluation revealed reduced motor function, mainly affecting the upper limbs. Muscle power at both upper limbs was graded 4/5 in all groups, except left elbow flexor, right wrist extensor and elbow extensor, which was a full 5/5. The lower limb muscle power was full except for bilateral hip flexion and right knee extension, where it was grade 4/5. Tendon reflexes were normal in both the upper and lower limbs. Sensory examination was normal, except for decreased pain sensation to pin-prick in both the upper and lower limb dermatomes.

Routine investigations that included hematological and biochemical investigations were normal. The chest radiograph was normal, and the cervical radiograph showed spondylotic changes. Ultrasonography of the abdomen and bilateral lower limbs showed no mass or lower limb deep vein thrombosis.

Whole-spine magnetic resonance imaging (MRI) with gadolinium was executed, and apart from degenerative changes of the spine, the MRI revealed several intradural extramedullary lesions at the cervical and thoracic spine, namely at C4–C5, T11, T12, and T12/L1 levels (Figure 1). These lesions demonstrated intermediate signal on T1-weighted images (WI), hyperintense signal on T2WI, and avid enhancement post contrast. The largest lesion was at the C4–C5 level, measuring at 1.0 × 2.7 cm, causing spinal cord compression with associated T2WI high-signal intensity within the cord-indicating cord edema (Figure 2). There was also mild crowding of the cauda equina at T12/L1 level. Multilevel abnormal marrow signal intensities were also noted in the T2, T9, and T10 vertebral bodies. The T9 vertebral body lesion was typical of a hemangioma, which showed hyperintense signal on T1WI, T2WI, and STIR sequences, with signal suppression on fat saturation (FS) sequences and minimal enhancement seen on post gadolinium sequences. However, the T2 and T10 lesions demonstrated a hyperintense signal on T2WI/STIR and were isointense on T1WI, with enhancement seen on the post-contrast sequence (Figure 3). In view of the multiplicity and characteristics of the spinal and osseous lesions and the background history of previous colon carcinoma, the diagnosis of leptomeningeal carcinomatosis with osseous metastases was made.

Owing to the presence of cervical cord compression, which was causing progressive clinical symptoms, he was subjected to a C4–C5 laminectomy accomplished via a posterior approach. A dural opening revealed a firm tumor anterior to the dentate ligament, arising from the right C4 nerve rootlet. The tumor was excised in total, and the involved nerve rootlet was sacrificed. Intraoperative diagnoses of neurofibroma or schwannoma were then made.

The postoperative condition was uneventful, with improvement of the upper limb power and intact upper and lower limb sensation. He was able to walk upon discharge. Contrast-enhanced computed tomography (CT) of the thorax, abdomen, and pelvis were carried out, showing degenerative changes at the cervical and lumbar spines as well as a well-defined lytic lesion with the corduroy sign at the T9 vertebral body suggestive of intraosseous hemangioma, consistent with the MR characteristics that are typical of hemangiomas. The other vertebral lesions seen within T2 and T10 were not appreciated on CT. There is no evidence to suggest recurrence of the colon carcinoma or distant metastasis.

Microscopic examination of the specimen showed neoplastic cells with a biphasic pattern, composed of mainly compact hypercellular Antoni A areas and occasional myxoid hypocellular Antoni B areas. The cells were spindle-shaped, hyperchromatic nuclei, some with cytoplasmic nuclear inclusions, and inconspicuous nucleoli with ill-defined eosinophilic cytoplasm. Nuclear palisading (Verocay bodies) was easily seen within the hypercellular areas. There were also collections of hemosiderin-laden macrophages and hyalinized blood vessels seen. No necrosis or mitosis. There is no evidence of malignancy. Final histopathological features were consistent with schwannoma.

As there were other intraspinal and bony lesions, not typical for vertebral hemangioma, seen at T2 and T10 vertebrae, a ^18^F-fluorodeoxyglucose positron emission tomography-computed tomography (^18^F-FDG PET/CT) was performed upon follow-up to exclude metastasis, which revealed reduced FDG metabolism of the T9 vertebral body lesion corresponding to a hemangioma (Figure 3f,g). The rest of the intradural, extramedullary, and osseous lesions, revealed no hypermetabolic uptake to suggest metastasis or local recurrence of colorectal carcinoma. Thus, these lesions were then concluded to be possible schwannomas and multiple lipid poor hemangiomas, respectively.

Discussion: Colorectal cancer is one of the most common cancers worldwide in terms of incidence and mortality, with an increasing trend, particularly in developing countries. Approximately 30% to 50% of newly diagnosed colorectal cases will progress into metastatic ones [1].

The presence of bone lesions in patients with a known primary malignancy usually points towards bone metastases, until and unless proven otherwise. Various imaging modalities are employed when assessing a bone lesion. These imaging modalities include both radiological imaging [e.g., plain radiographs, computed tomography (CT), and magnetic resonance imaging (MRI) as well as nuclear medicine imaging (e.g., bone scintigraphy, SPECT-CT, and PET-CT)]. Despite extensive imaging, diagnostic dilemma occasionally persists regarding the true nature of some bony lesions [2].

Intradural extramedullary (IDEM) tumors are the most commonly observed intraspinal tumors, comprising over 60% of tumors found within the spinal canal [3]. IDEM consists of a heterogeneous group of pathological entities, the vast majority of these lesions belong to one of three subtypes: meningiomas, schwannomas, or neurofibromas [4]. Other less commonly observed IDEM tumors include metastases, lipomas, paragangliomas, sarcomas, spinal nerve sheath myxomas, and vascular tumors [5]. In adults, IDEM metastases are much less common than extradural metastases and most commonly manifest as leptomeningeal disease. Conversely, intradural metastases are more common in children than extradural metastases, a reflection of the higher proportion of brain tumors with a propensity to spread by means of the cerebrospinal fluid (CSF) and the relative rarity of malignant cancers predisposed to bone metastasis within this population compared with adults. In adults, these tumors are commonly the result of drop lesions of intracranial metastasis from adenocarcinomas of the lung, prostate cancer, breast cancer, melanoma, or lymphomas [6,7].

The commong differentials for IDEM consists of one of three subtypes: meningiomas, schwannomas, or neurofibromas [4]. Meningioma is defined as a meningothelial (arachnoidal)-derived intracranial tumor. It is morphologically a heterogenous disease with a wide spectrum of appearances, as described in the latest WHO classification, and consists of 15 different histological subtypes [8]. Schwannoma, a slow growing benign tumor of Schwann cells, presents as a well encapsulated lesion. It is commonly seen along or attached to the peripheral, cranial, or sympathetic nerves, except in the optic and olfactory nerves, which lack the Schwann cell sheath. The specific details of their genesis are unknown [9]. Neurofibromas are benign, slow growing tumors that arise from the peripheral nerve sheath. Such tumors are comprised of a mixture of Schwann cells, perineurial cells, and fibroblast-like cells [10].

IDEM metastases of non-neurogenic origin are rare, representing less than 4% of all spinal metastases [11,12]. To our knowledge, only three such lesions have metastasized from a colorectal primary [6,13,14]. Rogers et al. were the first to report on an IDEM metastatic tumor in 1958 [15]. Since then, only 102 cases from various primaries have been reported [16]. In this case, even the primary malignancy was of non-CNS origin due to the multiplicity of the IDEM lesions with multiple suspicious osseous lesions; the MRI of the spine was interpreted as leptomeningeal nodules with osseous metastases. This was also consistent with the patient’s neurological symptoms of having progressive acute neurological deficits, which imply more aggressive pathology than benign disease that usually presents with non-progressive symptoms. However, the surveillance CT of this patient shows no evidence to suggest recurrence of the colorectal cancer. Moreover, it is less likely to have distant metastases, without systemic recurrence, especially since his disease has been in remission for a few years.

While IDEM metastases can occur at any level, the majority appear to occur in the lumbar spine. Lower CSF flow velocities in the lumbar spine may predispose this region to a higher probability of tumor cell settling [14,17]. This is in contrast to our patient, in whom the IDEM lesions were located at the cervical and thoracolumbar regions, with the largest being at the cervical region and causing cord compression. Careful review of the radiographic features of spinal cord tumors and other imaging may, at least in some cases, help differentiate metastatic disease from a second primary tumor. For example, in this case, retrospectively, its location at the cervical region could have suggested benign nerve sheath tumor, with the commonest benign tumor to occur at cervical spine being schwannoma [18]. However, what was against schwannoma was that it usually presents as a solitary tumor, especially in the elderly without a history of neurofibromatosis. Although the MRI of the spine was initially interpreted as leptomeningeal carcinomatosis, in retrospect, there was a subtle clue that could have supported the differential diagnosis of schwannoma or neurofibroma, although these two lesions have indistinguishable MRI characteristics. The clue that we can appreciate in the cervical region is the mild extension of the tumor into the right C4 neural foramina that is usually seen in a nerve sheath tumor.

The other diagnostic dilemma in this case was the presence of multiple osseous lesions. Although one of the osseous lesions showed typical hemangioma features, the rest of these lesions were indeterminate. The stereotypical hemangioma usually causes no problems because, on MRI, it appears hyperintense on T1WI and T2WI and enhances intensely on post contrast T1WI fat-suppressed images. These lesions are also usually clearly demarcated. However, some hemangiomas do not have these stereotypical characteristics. Atypical hemangiomas, which may vary in appearance, include those that are hypointense on T1WI but retain the typical characteristics on T2-weighted and fat-suppressed postcontrast images. Thus, these atypical features of hemangiomas cause difficulty in distinguishing benign lesions from metastases [19].

It is possible that newer imaging modalities, such as positron emission tomography, may also prove helpful in determining the nature of spinal cord tumors [20]. Similar to our case, ^18^F-FDG PET/CT, which was performed upon follow-up, revealed reduced FDG metabolism of the T9 vertebral body lesion corresponding to a hemangioma. The rest of the spinal lesion revealed no hypermetabolic uptake to suggest metastasis. Thus, these small lesions at the T2 and T10 vertebral bodies may also represent atypical hemangiomas. The rest of the IDEM lesions also showed no hypermetabolic uptake to suggest metastasis. These characteristics, together with histopathological findings, led to the final diagnosis of multiple schwannoma.

This case underscores the need to consider the diagnosis of sporadic benign IDEM lesions when evaluating patients with a known primary with multiple imaging abnormalities. IDEM metastasis is rare, even in patients with a known previous primary. The multiplicity of the lesions with multiple underlying bone lesions had imposed a diagnostic challenge in this case. Thus, a thorough and careful review of radiographic features with the detection of subtle imaging findings may be helpful to differentiate metastatic disease from a second primary tumor.

## Figures and Tables

**Figure 1 diagnostics-13-01254-f001:**
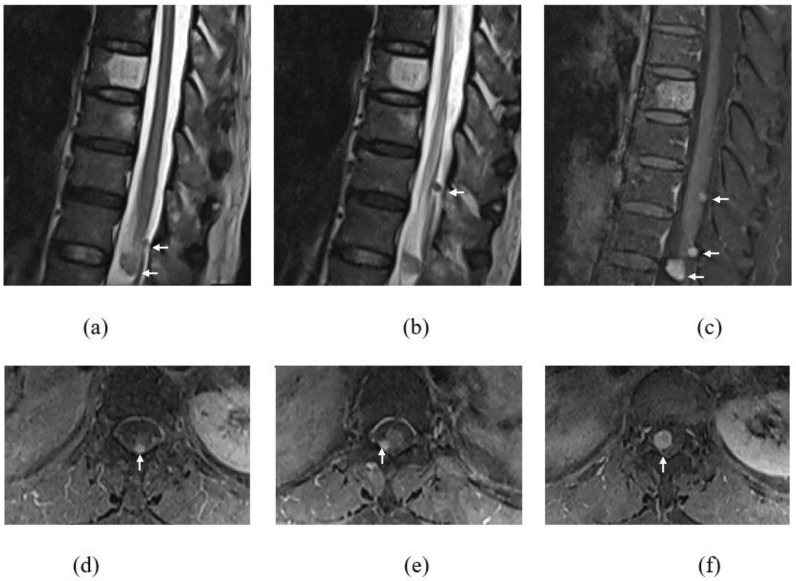
T2WI (**a**,**b**) sagittal images of thoracic and upper lumbar spine showing multiple (white arrows) intradural extramedullary lesions at T11, T12 and T12/L1 levels. Gadolinum-T1FS sequence in sagittal (**c**) and axial (**d**–**f**) planes showing enhancement of all these lesions.

**Figure 2 diagnostics-13-01254-f002:**
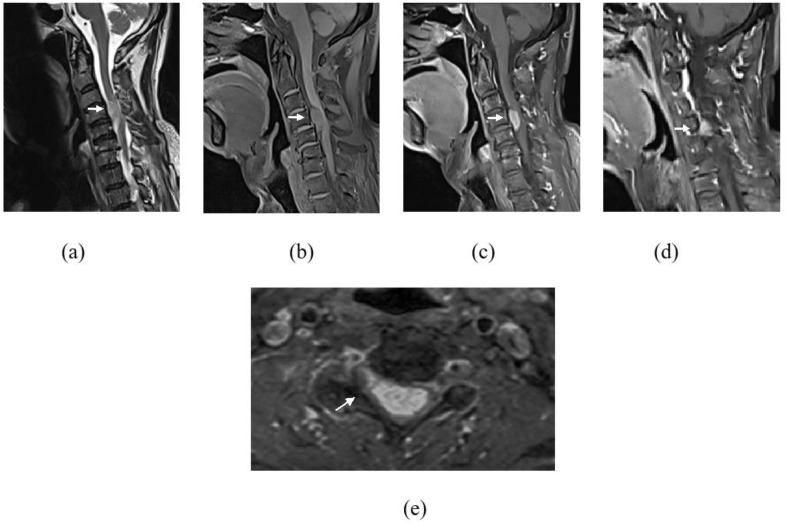
Images of cervical spine showing an intradural extramedullary lesion (white arrow) at the level of C4–C5. It demonstrates hyperintense signal on T2WI (**a**), intermediate signal on T1WI (**b**) and avid enhancement on gadolinum-T1FS sequence (**c**,**e**). This lesion causes spinal cord compression at this level with associated spinal cord T2WI hyperintense signal indicating of cord oedema (**a**). Right parasagittal image (**d**) showed extension into the right C4 neural foramina (**e**).

**Figure 3 diagnostics-13-01254-f003:**
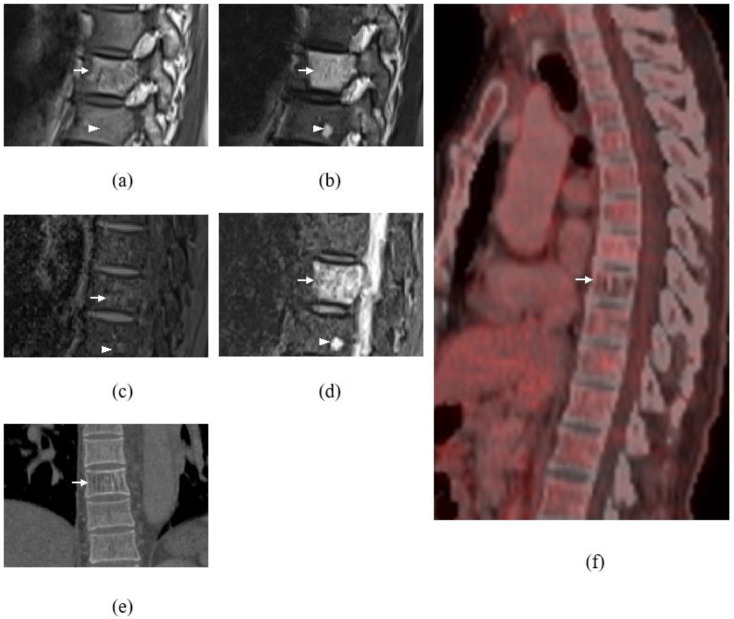
Sagittal images (**a**–**d**) at T8 to T10 level showing lesion (white arrow) at T9 vertebral body demonstrating hyperintense signal on T1WI (**a**) and T2WI (**b**) with signal suppression on fat saturation sequence (**c**) and minimal enhancement on post contrast sequence (**d**), typical of hemangioma. Coronal bone algorithm CT shows classic corduroy sign in hemangioma at T9 vertebral body (**e**). Sagittal plane of thoracolumbar spine (f) of ^18^F-FDG PET/CT also showed reduced FDG metabolism of the T9 lesion (white arrow) with no FDG avid lesion seen at the thoracolumbar spine to suggest metastasis. Another small lesion (arrowhead) at T10 vertebral body which is isointense on T1WI (**a**), hyperintense on T2WI (**b**) with enhancement on post contrast sequence (**d**).

## Data Availability

Not applicable.

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
