# Peer review of "Spinal Schwannomatosis Mimicking Metastatic Extramedullary Spinal Tumor"

_diagnostics, 2023, doi:10.3390/diagnostics13071254_

Round 1

Reviewer 1 Report

Very interesting case well presented and well described by the authors.

Significance of content - the paper presents very interesting case which very often creates a lot of diagnostic difficulties in routine practice when patients with history of oncological disease present multiple diffuse lesions in organs which may be also common organs of metastases. Thus it is clinically significant issue and may be very helpful in clinical practice.

Quality of presentation - the authors described in a very clear manner the history of the patient, diagnostic dificulties and diagnostic algorithm which was followed. In my opinion it is very valuable pathway to share. Moreover the case was supported by very good quality images, which makes the presentation of the case complete.

Scientific soundness - the paper describes very important case of coexistance of multiple hemangiomas and multiple schwannomas in a patient with a history of carcinoma, which is not a very common combination thus in my opinion it should be published.

Author Response

Dear Reviewer,
We thank you for your kind comments and generous feedback.

Regards, Authors

Reviewer 2 Report

Thank you for the opportunity to review this manuscript.

The authors provide a case report on spinal schwannomatosis mimicking metastatic extramedullary spinal tumor manifestation in a patient with a history of colon cancer.

Layout and format: The manuscript is structured and meets the target journal's expected format. 

Title: The title of the manuscript does not reflect the article's content.
Abstract: The abstract is well-structured and reflects the content of the article. 
Introduction: The introduction describes the aim of the study accurately. 
Methods: The authors describe data acquisition and differential diagnosis and provide several radiographs.
Results: The presentation of the results is clear and stringent. The significant limitations are not listed.

-The authors present a compelling case meticulously, which is rare but of interest to the community.

-The manuscript contains a case report without a proper literature review. In my view, the title should mention the nature of the article, and before reconsideration for publication in a scientific journal, a review of the literature should be performed. Thus, the outline of the manuscript should be changed accordingly.

-Extensive language editing is mandatory.

My recommendation: Major revision.

Author Response

Dear Reviewer,
We thank you for your kind comments and feedback.

  1. Title: The title of the manuscript has been changed to reflect the article's content.
  2. Discussion and literature review has been improved as suggested by the reviewer.
  3. English language check has been done.

We thank the reviewers again for their generous input.

Round 2

Reviewer 2 Report

Thank you for the opportunity to review the revised manuscript.

In my opinion, the authors adequately addressed the concerns raised by my fellow reviewer.